# On the Trade-off of Intra-/Inter-class Diversity for Supervised Pre-training

**Jieyu Zhang**[1]*, **Bohan Wang**[2]*, **Zhengyu Hu**[3], **Pang Wei Koh**[1], **Alexander Ratner**[1,4]

[1] University of Washington    [2] USTC    [3] HKUST(GZ)    [4] Snorkel AI, Inc.

`{jieyuz2,pangwei,ajratner}@cs.washington.edu`
`bhwangfy@gmail.com`
`zhu021@connect.hkust-gz.edu.cn`

## Abstract

Pre-training datasets are critical for building state-of-the-art machine learning models, motivating rigorous study on their impact on downstream tasks. In this work, we study the impact of the trade-off between the intra-class diversity (the number of samples per class) and the inter-class diversity (the number of classes) of a supervised pre-training dataset. Empirically, given a fixed pre-training dataset size, we find that the best downstream performance comes with a balance on the intra-/inter-class diversity. To understand the underlying mechanism, we show theoretically that downstream performance depends monotonically on both types of diversity. Notably, our theory reveals that the optimal class-to-sample ratio ($\frac{\text{\#classes}}{\text{\#samples per class}}$), *i.e.*, the ratio of the number of pre-training classes to the number of samples per class, is invariant to the size of the pre-training dataset, enabling the prediction of the optimal number of pre-training classes. We demonstrate the effectiveness of this application by an improvement of approximately 2 points on average on downstream tasks when pre-training on ImageNet.

## 1 Introduction

Many state-of-the-art deep neural network models are pre-trained on large datasets before being finetuned for downstream tasks [13, 17, 23, 1]. While the composition of their pre-training dataset has been shown to be a key factor in the performance of these models [7, 9, 14, 8, 12, 26], how best to design these pre-training datasets still remains underexplored. In this work, we focus on supervised pre-training, one of the most popular pre-training paradigms, and study two key quantities of a supervised pre-training dataset: intra-class diversity (the number of different samples within each pre-training class) and inter-class diversity (the number of different pre-training classes). Intuitively, both diversities are beneficial for supervised pre-training [13]. Yet when the size of the pre-training dataset is fixed, these diversities trade off, since increasing one will decrease the other. Our work studies the impact of this dataset diversity trade-off on downstream performance, as well as how to balance them to design a supervised pre-training dataset with the best downstream performance.

Empirically, with ImageNet [24] as the pre-training dataset and the pre-training dataset size fixed, we show that the optimal performance on the downstream tasks occurs when a balance on the intra-/inter-class diversity is achieved. We then offer a theoretical explanation for this effect by first modeling the dataset generation process through a two-step sampling framework, and then demonstrating that the test error of the downstream task displays a rational relationship with respect to the class-to-sample ratio, *i.e.*, the ratio of the number of pre-training classes to the number of samples per class, or, in other words, the ratio between inter-/intra-class diversity. The established analytical relationship

---

*These authors contributed equally to this work.

between downstream performance and the class-to-sample ratio can serve as a guiding principle in designing a supervised pre-training dataset by estimating the optimal class-to-sample ratio rather than the grid search.

Notably, our theory shows that given a source of a pre-training dataset and a downstream task, the optimal class-to-sample ratio is invariant to the size of the pre-training dataset. Based on such an invariance, one could estimate the optimal class-to-sample ratio with small pre-training datasets and then leverage it to build a large-scale pre-training dataset. In particular, the optimal number of pre-training classes $\bar{K}$ and the number of examples per class $n$ are proportional to the square root of the size of the pre-training dataset $N$, $i.e.$, $\bar{K} \propto \sqrt{N}$, which leads to an invariant optimal class-to-sample ratio. We empirically verify our theoretical findings on ImageNet [24] and present the effectiveness of its application in predicting the optimal number of classes for pre-training datasets with different sizes. In addition, we conducted experiments with different pre-trained datasets, different model backbones, and downstream tasks of different domains to demonstrate that our findings are consistent across many scenarios.

Our major findings and contributions are as follows:

- In supervised pre-training, we observe that with a fixed pre-training dataset size, there exists a trade-off between intra-class and inter-class diversities. This balance between diversities plays a crucial role in shaping the downstream performance, underscoring the significance of considering both aspects when designing the pre-training dataset;

- We then theoretically explain this effect by first modeling the dataset generation process through a two-step sampling framework and then showing that the test error of the downstream task displays a convex relationship with respect to the class-to-sample ratio, serving as a guiding principle in designing a supervised pre-training dataset.;

- Our theory also uncovers the invariance of the optimal class-to-sample ratio with respect to the size of the pre-training dataset, allowing us to predict the optimal number of classes with a small number of pre-training data before building a larger pre-training dataset for a downstream task.

## 2   Empirical Observations

The goal of this work is to study the trade-off of intra-/inter-class diversity in a supervised pre-training dataset and its impact on the pre-trained model's performance on downstream tasks. Specifically, the inter-class diversity refers to the diversity of classes in pre-training dataset, $i.e.$, how many different classes we have ($K$); while the intra-class diversity refers to the diversity of samples within each class, $i.e.$, how many different samples in each class ($n$). When the size of the pre-training dataset is fixed, increasing either type of diversity will by definition decrease the other, leading to a dataset diversity trade-off. To study the impact of such dataset diversity trade-off, we experiment with pre-training datasets with varying numbers of classes and number of samples per class. The experimental details can be found in Appendix A.2.

**Evaluation protocol.** Following common practice [13], we use the ImageNet [24] as the dataset for supervised pre-training. In this work, we mainly use ResNet-18 [10] as the backbone model. For evaluating the performance of the pre-trained model on downstream tasks, we perform linear probing (tuning the head but freezing the lower layers). We repeat each individual experiment five times and report the averaged top-1 accuracy.

**Downstream tasks.** We adopt the following six datasets as the downstream classification tasks: Stanford40 dataset [25] for action recognition, StanfordDogs [15] for fine-grained object recognition, MIT67 [22] for scene classification, CIFAR10 [16] for image recognition datasets, Flowers102 [20] for image classification dataset, FGVCAircraft [18] for aircraft classification dataset.

While having all the other configurations fixed, during pre-training, we vary the number of classes and the number of samples per class. Specifically, given $N$ as the size of the pre-training dataset and $K$ as the number of classes, we randomly sample $K$ classes from ImageNet and then uniformly sample $n = \frac{N}{K}$ samples from each class to compose the dataset. We experiment with the following $N$ and $K$ values: {1K, 2K, 5K, 10K, 20K, 50K, 100K} and {2, 5, 10, 20, 50, 100, 200, 500, 1000} respectively. Note that with larger $N$ ($e.g.$, $N = 10K$), we cannot evaluate smaller values of $K$ ($e.g.$, $K = 2$), since in ImageNet each class has at most 1300 samples.

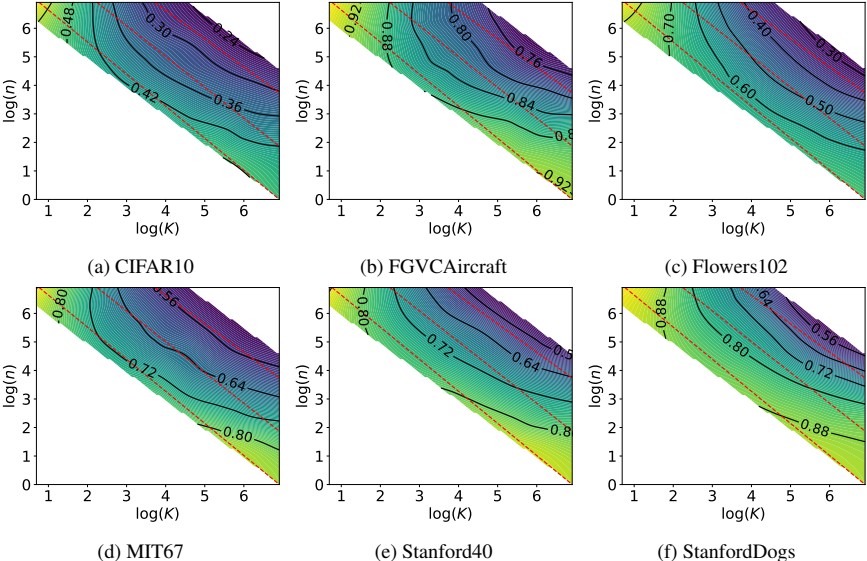

Figure 1: Test error rate (the darker, the better) as a function of intra-class diversity on the y-axis ($\log n$) and inter-class diversity on the x-axis ($\log K$). Each plot represents a different dataset. The red dashed anti-diagonal lines indicate fixed pre-training dataset sizes ($\log K + \log n = \log N$ is constant), from which we can see that obtaining the best downstream performance given a fixed pre-training dataset size requires balancing both diversities.

**Results and observations.** We visualize the results in Figure 1. In the contour plot, the z-value is the error rate on the test set, thus lower is better. The x-axis and y-axis are inter-class diversity ($\log K$) and intra-class diversity ($\log n$) in the log space, respectively. The values on anti-diagonal lines ($y = -x + c$) share the same pre-training dataset size as $\log N = \log K + \log n$. From the results, we have two observations: 1) **Both intra-/inter-class diversity are beneficial for downstream tasks:** We can see that increasing either inter-class diversity (($\log K$) or intra-class diversity ($\log n$), given the other is fixed, would lead to a better test error rate. This is intuitive and as expected, since the size of the pre-training dataset $N$ would increase accordingly, which is known to be beneficial for downstream tasks [11, 7, 13]. 2) **A trade-off of intra-/inter-class diversity on downstream task performance:** More importantly, by looking at the anti-diagonal lines where $\log N$ is fixed and equals $\log K + \log n$, we can see a trade-off between intra-/inter-class diversity on the test error rate of downstream tasks: either cases of 1) high inter-class diversity, low intra-class diversity and 2) low inter-class diversity and high intra-class diversity would not render the best performance. Instead, some point in the middle of the anti-diagonal line leads to the lowest test error rate.

## 3 Theoretical Understanding

In this section, we first present the theoretical setup and notations and then provide a theory on the impact of the pre-training dataset diversity on downstream performance. We also show that the optimal class-to-sample ratio ($\frac{K}{n}$) is invariant to the size of the pre-training dataset; such a property can be leveraged to predict the optimal number of classes for building a large pre-training dataset with a small number of data samples first.

### 3.1 Setup and notations

**Dataset.** To be consistent with our experimental setup, we consider the supervised pre-training task. Specifically, we can access two datasets, one for the pre-training task (denoted as $S^p$) and another for the downstream task (denoted as $S^d$). Each example in the *pre-training* dataset consists of input features $x \in \mathcal{X} = \mathbb{R}^{d_1}$ (where $d_1$ is the dimension of data) and a label $y \in [K]$ (where $K$ is the number of classes). Specifically, we denote $S^p = \{(x_1, y_1), \cdots, (x_N, y_N)\}$, where $N$ is the size of $S^p$, and assume that $S^p$ is sampled according to some underlying distribution $\mathcal{P}$ (we do

not specify $\mathcal{P}$ here because we will analyze cases with different $\mathcal{P}$ latter). Every example in the *downstream* dataset consists of input features $\tilde{x} \in \mathcal{X}$ and a label $\tilde{y} \in [\tilde{K}]$ (note that $\tilde{K}$ does not necessarily equal-to $K$), and is sampled *i.i.d.* according to an underlying distribution $\tilde{\mathcal{P}}$. We denote $S^d = \{(\tilde{x}_1, \tilde{y}_1), \cdots, (\tilde{x}_N, \tilde{y}_{\tilde{N}})\}$ and thus $S^d \sim \tilde{\mathcal{P}}^{\tilde{N}}$.

**Model.** The models for both pre-training and downstream tasks consist of two components: the feature extractor and the classifier. Specifically, the model for the pre-training task is given as $f_{S^p} \circ h_{S^p}$, where $f_{S^p} : \mathbb{R}^{d_2} \to \mathbb{R}^K$ is the pre-training classifier ($d_2$ is the dimension of feature) and $h_{S^p} : \mathbb{R}^{d_1} \to \mathbb{R}^{d_2}$ is the feature extractor. We denote the set of all possible $f_{S^p}$ as $\mathcal{F}$, and the set of all possible $h$ as $\mathcal{H}$. The model for the downstream task is given as $f_{S^d} \circ h_{S^d}$, where $f_{S^d} : \mathbb{R}^{d_2} \to \mathbb{R}^{\tilde{K}}$ is the downstream classifier and $h$ is the feature extractor shared with the pre-training task. We set all possible $f_{S^p}$ as $\tilde{\mathcal{F}}$.

**Loss.** To measure the correctness of model's predictions, we use the cross-entropy loss. Specifically, given an example $(x, y)$ and pre-training/downstream model $f \circ h$, the corresponding cross-entropy loss is defined as

$$\ell(f \circ h(x), y) = -\log \frac{e^{f_y \circ h(x)}}{\sum_i e^{f_i \circ h(x)}},$$

where $f_i \circ h(x)$ is the $i$-th coordinate of $f \circ h(x)$. We make the following assumption about the complexity of the model.

**Assumption 1.** *There exist a positive constant $M_\ell$, such that $\forall i$, $\forall f \in \mathcal{F} \cup \tilde{\mathcal{F}}, h \in \mathcal{H}, x \in \mathcal{X}$,*

$$\ell(f \circ h(x), i) \leq M_\ell.$$

*Furthermore, for any distribution $Q$ over the feature space $\mathbb{R}^{d_1}$, any $m \in \mathbb{N}$, and any $\mathcal{F}' \in \{\mathcal{F}, \tilde{\mathcal{F}}\}$, define the Gaussian complexity of function class $\mathcal{F}' \circ \mathcal{H}$ over the marginal distribution $Q$ with sample number $m$ as*

$$G_m^Q(\mathcal{F}' \circ \mathcal{H}) \triangleq \mathbb{E}_{(x_i)_{i=1}^m \sim Q^m} \mathbb{E}_{(\sigma_{i,j})_{i \in [N], j \in K'} \sim \mathcal{N}(0, \mathbb{1}_{n \times K'})} \sup_{f \in \mathcal{F}, h \in \mathcal{H}} \sum_{i=1}^m \sum_{j=1}^{K'} \sigma_{i,j} f_j \circ h(x_i).$$

*Here $K' = \dim(\mathcal{F}')$. We assume that $G_m^Q(\mathcal{F} \circ \mathcal{H}) \leq G\sqrt{m}$, where $G$ is independent of $Q$ and $m$.* [2]

Assumption 1 constrains the dependence of model complexity over the number of sample $N$, which is a standard assumption in generalization analysis [19].

**Risks and corresponding minimizers.** We first define the empirical risk $\bar{\mathcal{R}}_p(f \circ h, S^p)$ and population risk $\mathcal{R}_p(f \circ h, \mathcal{P})$ over the pre-training task as

$$\bar{\mathcal{R}}_p(f \circ h, S^p) \triangleq \frac{1}{N} \sum_{i=1}^N [\ell(f \circ h(x_i), y_i)], \mathcal{R}_p(f \circ h, \mathcal{P}) \triangleq \mathbb{E}_{S^p \sim \mathcal{P}} \bar{\mathcal{R}}_p(f \circ h, S^p).$$

The corresponding feature extractor and classifier of the empirical risk minimizer over the pre-training task as

$$h_{S^p} \triangleq \arg\min_{h \in \mathcal{H}} \left( \min_{f \in \mathcal{F}} \bar{\mathcal{R}}_p(f \circ h, S^p) \right), f_{S^p} \triangleq \arg\min_{f \in \mathcal{F}} \left( \min_{h \in \mathcal{H}} \bar{\mathcal{R}}_p(f \circ h, S^p) \right).$$

Given a feature extractor $h \in \mathcal{H}$, we measure its performance over the pre-training task through a classifier agnostic approach by considering

$$\mathcal{R}_p(h, \mathcal{P}) \triangleq \min_{f \in \mathcal{F}} \mathcal{R}_p(f \circ h, \mathcal{P}).$$

Note here we slightly abuse the notation of $\mathcal{R}_p$ without causing confusion as $f \circ h$ and $h$ have different image spaces. The representation error of $h$ over $\mathcal{P}$ is then defined as the gap between the risk of $h$ and the smallest possible risk

$$\mathcal{E}_p(h, \mathcal{P}) \triangleq \mathcal{R}_p(h, \mathcal{P}) - \min_{\tilde{h} \in \mathcal{H}} \mathcal{R}_p(\tilde{h}, \mathcal{P}).$$

---

[2]This inequality holds for a wide range of models, including deep neural networks [2].

Similarly, the empirical risk $\bar{\mathcal{R}}_d(f \circ h, S^d)$ and population risk $\mathcal{R}_d(f \circ h, \tilde{\mathcal{P}})$ over the downstream task are defined as

$$\bar{\mathcal{R}}_d(f \circ h, S^d) \triangleq \sum_{i=1}^{\tilde{N}} \ell(\tilde{f} \circ h(\tilde{x}_i), \tilde{y}_i),\ \mathcal{R}_d(f \circ h, \tilde{\mathcal{P}}) \triangleq \mathbb{E}_{S^d \sim \tilde{\mathcal{P}}^{\tilde{N}}} \bar{\mathcal{R}}_d(f \circ h, S^d).$$

The learned classifier from the downstream task can then be defined as

$$f_{S^d} \triangleq \arg\min_{f \in \tilde{\mathcal{F}}} \left( \min_{h \in \mathcal{H}} \bar{\mathcal{R}}_d(f \circ h_{S^p}, S^d) \right).$$

The corresponding performance and excess risk of $h$ over the downstream task can then be defined as

$$\mathcal{R}_d(h, \tilde{\mathcal{P}}) \triangleq \min_{\tilde{f} \in \tilde{\mathcal{F}}} \mathcal{R}_d(\tilde{f} \circ h, \tilde{\mathcal{P}}),\ \mathcal{E}_d(h, \tilde{\mathcal{P}}) \triangleq \mathcal{R}_d(h, \tilde{\mathcal{P}}) - \min_{\tilde{h} \in \mathcal{H}} \mathcal{R}_d(\tilde{h}, \tilde{\mathcal{P}}).$$

Finally, we are interested in the excess risk of the obtained model $f_{S^d} \circ h_{S^p}$ over the downstream task, i.e.,

$$\mathcal{E}_d(f_{S^d} \circ h_{S_p}, \tilde{\mathcal{P}}) \triangleq \mathcal{R}_d(f_{S^d} \circ h_{S_p}, \tilde{\mathcal{P}}) - \min_{\tilde{h} \in \mathcal{H}} \mathcal{R}_d(\tilde{h}, \tilde{\mathcal{P}}).$$

### 3.2 A theory on the impact of intra-/inter-class diversity trade-off

In this work, we focus on a common practice of collecting the pre-training dataset: one first sample $K$ classes and then collect $n$ samples for each class. We start with a detailed characterization of such data generation process, followed by assumptions and the main result.

**Data generation process 1.** We assume the pre-training data is generated through a two-step sampling process. Specifically, suppose that there is a distribution $\mathcal{D}$ over $\Delta(\mathcal{X})$ (the set consisting of all distributions on $\mathcal{X}$). Then, $S^p$ is generated by the following procedure (and $\mathcal{P}$ is naturally induced)

- Sample $K$ classes by i.i.d. sampling $K$ distributions $\{\mathcal{P}_i\}_{i=1}^K$ according to $\mathcal{D}$. These are respectively the underlying distributions of $K$ classes;
- For each $i \in [K]$, i.i.d. sample $n$ data $\{x_{i,1}, \cdots, x_{i,n}\}$ according to $\mathcal{P}_i$ and denote $S_i = \{(x_{i,1}, i), \cdots, (x_{i,n}, i)\}$. Note here $x_{i,j}$ does not contain the information of label, as its label information is already contained in $i$. The whole dataset is obtained by putting all $S_i$ together, i.e., $S^p = \{S_1, \cdots, S_K\}$.

We make the following assumption on the correlation between the representation powers of the pre-training and the downstream task.

**Assumption 2.** *Given $\mathcal{P}$, there exists non-negative coefficients $\nu_0^{\tilde{\mathcal{P}}}(\mathcal{P})$ and $\nu_1^{\tilde{\mathcal{P}}}(\mathcal{P})$, such that $\forall h \in \mathcal{H}$,*

$$\mathcal{E}_d(h, \tilde{\mathcal{P}}) \leq \nu_1^{\tilde{\mathcal{P}}}(\mathcal{P}) \mathcal{E}_p(h, \mathcal{P}) + \nu_0^{\tilde{\mathcal{P}}}(\mathcal{P}).$$

*We further assume that $\nu_0^{\tilde{\mathcal{P}}}(\mathcal{P})$ and $\nu_1^{\tilde{\mathcal{P}}}(\mathcal{P})$ are stable, that is, there exist two $\tilde{\mathcal{P}}$-dependent positive constants $M_0^{\tilde{\mathcal{P}}}$ and $M_1^{\tilde{\mathcal{P}}}$, such that for any $\mathcal{P} = \Pi_{i=1}^K (\mathcal{P}_i, i)^n$ and $\mathcal{P}' = \Pi_{i=1}^{K-1}(\mathcal{P}_i, i)^n \times (\mathcal{P}'_K, K)^n$ which differ by only one component, we have that $|\nu_0^{\tilde{\mathcal{P}}}(\mathcal{P}) - \nu_0^{\tilde{\mathcal{P}}}(\mathcal{P}')| \leq \frac{M_0^{\tilde{\mathcal{P}}}}{K}$ and $|\nu_1^{\tilde{\mathcal{P}}}(\mathcal{P}) - \nu_1^{\tilde{\mathcal{P}}}(\mathcal{P}')| \leq \frac{M_1^{\tilde{\mathcal{P}}}}{K}$. Moreover, we assume that $\nu_0^{\tilde{\mathcal{P}}}(\mathcal{P})$ and $\nu_1^{\tilde{\mathcal{P}}}(\mathcal{P})$ concentrate around their means, i.e., there exist $\nu_0^{\tilde{\mathcal{P}}}(\mathcal{D})$, $\nu_1^{\tilde{\mathcal{P}}}(\mathcal{D})$, $C_0^{\tilde{\mathcal{P}}}$ and $C_1^{\tilde{\mathcal{P}}}$, such that $|\mathbb{E}_{\{\mathcal{P}_i\}_{i=1}^K \sim \mathcal{D}^K} \nu_0^{\tilde{\mathcal{P}}}(\mathcal{P}) - \nu_0^{\tilde{\mathcal{P}}}(\mathcal{D})| \leq \frac{C_0^{\tilde{\mathcal{P}}}}{\sqrt{K}}$ and $|\mathbb{E}_{\{\mathcal{P}_i\}_{i=1}^K \sim \mathcal{D}^K} \nu_1^{\tilde{\mathcal{P}}}(\mathcal{P}) - \nu_1^{\tilde{\mathcal{P}}}(\mathcal{D})| \leq \frac{C_1^{\tilde{\mathcal{P}}}}{\sqrt{K}}.$*

Assumption 2 assumes that the pre-training representation error can bound the downstream representation error, which is a common assumption in existing works [6, 26, 3]. Also, as $\mathcal{P}$ is derived by sampling $K$ distributions according to $\mathcal{D}$, we make mild assumptions that the coefficients $\nu_0^{\tilde{\mathcal{P}}}(\mathcal{P})$ and $\nu_1^{\tilde{\mathcal{P}}}(\mathcal{P})$ is robust when changing the underlying distribution of only one class, and when $K$ grows, the expectation of $\nu_0^{\tilde{\mathcal{P}}}(\mathcal{P})$ and $\nu_1^{\tilde{\mathcal{P}}}(\mathcal{P})$ converge to some limits.

**Theorem 3.1.** *Let Assumptions 1 and 2 hold. For **data generation process 1**, with probability over the sampling of the datasets at least $1 - \delta$, we have*

$$\mathcal{E}_d(f_{S^d} \circ h_{S_p}, \tilde{\mathcal{P}}) \leq \left( \nu_1^{\tilde{P}}(\mathcal{D}) + M_1\sqrt{\frac{\log\frac{4}{\delta}}{2K}} + \frac{C_1}{\sqrt{K}} \right) \left( 5M_\ell\sqrt{\frac{\log\frac{6}{\delta}}{2n}} + \frac{2G\sqrt{2}}{\sqrt{n}} \right) + \nu_0^{\tilde{P}}(\mathcal{D})$$

$$+ M_0\sqrt{\frac{\log\frac{6}{\delta}}{2K}} + \frac{C_0}{\sqrt{K}} + 5M_\ell\sqrt{\frac{\log\frac{6}{\delta}}{2\tilde{N}}} + 2\sqrt{2}G\frac{1}{\sqrt{\tilde{N}}}. \tag{1}$$

The detailed proof is deferred to Appendix A.8.1. Below we simplify the right-hand-side of Equation 1 and show that the empirically observed downstream performance trade-off can be explained by such a result.

**Simplifying the Theorem 3.1.** Denote the RHS of Equation 1 as $U$, we have

$$U = \nu_1^{\tilde{P}}(\mathcal{D}) \left( 5M_\ell\sqrt{\frac{\log\frac{6}{\delta}}{2}} + 2G\sqrt{2} \right) \frac{1}{\sqrt{n}} + \left( M_0\sqrt{\frac{\log\frac{6}{\delta}}{2}} + C_0 \right) \frac{1}{\sqrt{K}} + \left( M_1\sqrt{\frac{\log\frac{6}{\delta}}{2}} + C_1 \right)$$

$$\times \left( 5M_\ell\sqrt{\frac{\log\frac{6}{\delta}}{2}} + 2G \right) \frac{1}{\sqrt{N}} + \nu_0^{\tilde{P}}(\mathcal{D}) + 5M_\ell\sqrt{\frac{\log\frac{6}{\delta}}{2\tilde{N}}} + 2\sqrt{2}G\frac{1}{\sqrt{\tilde{N}}}.$$

We can see that the above equation can be simplified as

$$U = \frac{A}{\sqrt{n}} + \frac{B}{\sqrt{K}} + \frac{C}{\sqrt{N}} + D, \tag{2}$$

where $A, B, C, D$ do not depend on $N, K, n$, but instead only depend on the properties of the underlying pre-training and the downstream task data distribution.

**Explaining downstream performance trade-off given a fixed $N$.** From Equation 2 we can see that the performance on the target task would increase when we increase 1) intra-class diversity $n$, 2) inter-class diversity $K$, and 3) the size of pre-training dataset $N$. When $N$ is fixed, however, increasing either intra-class diversity or inter-class diversity would decrease the other (since $N = n \times K$) and therefore eventually lead to a performance drop. Another way to see this is to parameterize $U$ as a function of $K$ without $n$:

$$U(K) = \frac{A\sqrt{K}}{\sqrt{N}} + \frac{B}{\sqrt{K}} + \frac{C}{\sqrt{N}} + D, \tag{3}$$

From this we can clearly see that both extremes of $K$ (too large or too small) would not lead to optimal performance. A similar conclusion can be drawn regarding $n$ when parametrizing $U$ as a function of $n$ without $K$.

### 3.3 Balancing intra-/inter-class diversity: the optimal class-to-sample ratio

When $N$ is fixed, by leveraging the fact that $N = n \times K$, we can express $U$ as

$$U = \frac{1}{N^{\frac{1}{4}}} \left( Ax^{\frac{1}{4}} + B\frac{1}{x^{\frac{1}{4}}} \right) + c, \tag{4}$$

where $c = \frac{C}{\sqrt{N}} + D$ is a constant and $x = \frac{K}{n}$ is the class-to-sample ratio. To minimize $U$, we have the optimal class-to-sample ratio $\bar{x} = \frac{B^2}{A^2}$. Notably, because both $A$ and $B$ have no dependency on $N$, **the optimal class-to-sample ratio for a specific downstream task is invariant to the size of the pre-training dataset.** Motivated by this, one could estimate the optimal class-to-sample ratio using a small $N$ and then use it to predict the optimal number of classes for building a large pre-training dataset. In particular, given the optimal class-to-sample ratio $\bar{x}$, the optimal number of classes is $\bar{K} = \frac{B}{A}\sqrt{N}$. Based on Equation 4, one only needs three (class-to-sample ratio, performance) tuples to estimate the constants $(A, B, c)$ with a fixed $N$ for computing the optimal class-to-sample ratio.

### 3.4 When no need to balance intra-/inter-class diversity: a contrasting case

So far, we have studied a common practice of collecting the pre-training dataset, *i.e.*, first sample $K$ classes and then collect $n$ samples for each class, and showed that when the size of the pre-training dataset $N$ is fixed, neither too large nor too small $K$ is optimal. However, there exist other ways of collecting the pre-training dataset. For example, one could collect a fixed set of data samples and then manipulate the value of $K$ by clustering the samples into any number of clusters. This raises a natural question: is our conclusion still valid for this case? To answer this question and further understand the condition for our theory, in this section, we study a contrasting case of the data generation process corresponding to the aforementioned example, with which the trade-off no longer exists and, instead, $K$ dominates the downstream performance. Concretely, the data generation process is given as follows:

**Data generation process 2.** The pre-training dataset is generated by first i.i.d. sampling data $\{x_1, x_2, \cdots, x_N\}$ according to some distribution $\mathcal{P}_{\mathcal{X}} \in \Delta(\mathcal{X})$. We then obtain the label $y_i \in [K]$ of each $x_i$ by performing clustering. The final pre-training dataset is given as $S = \{(x_1, y_1), \cdots, (x_N, y_N)\}$ (and $\mathcal{P}$ is naturally induced).

The new data generation process is different from the one in Section 3.2. In particular, with different $K$, the new data generation process would not introduce new data samples as we manipulate the label by clustering the current sampled data, while for the data generation process in Section 3.2, the sampled data change according to the classes we picked.

As an analogy to Assumption 2, we make the following assumption on the correlation between the representation powers of the pre-training and the downstream task and then present the theorem.

**Assumption 3.** *We assume the pre-training representation error can bound the downstream representation error. Specifically, there exists non-negative constants $\nu_0^{\tilde{\mathcal{P}}}(\mathcal{P})$ and $\nu_1^{\tilde{\mathcal{P}}}(\mathcal{P})$, such that $\forall h \in \mathcal{H}$,*

$$\mathcal{E}_d(h, \mathcal{P}) \leq \nu_1^{\tilde{\mathcal{P}}}(\mathcal{P})\mathcal{E}_p(h) + \nu_0^{\tilde{\mathcal{P}}}(\mathcal{P}).$$

*We further assume that $\nu_0^{\tilde{\mathcal{P}}}(\mathcal{P})$ and $\nu_1^{\tilde{\mathcal{P}}}(\mathcal{P})$ concentrate to their means, i.e., there exist $\nu_0^{\tilde{\mathcal{P}}}(\mathcal{P}_{\mathcal{X}})$, $\nu_1^{\tilde{\mathcal{P}}}(\mathcal{P}_{\mathcal{X}})$, $C_0^{\tilde{\mathcal{P}}}$ and $C_1^{\tilde{\mathcal{P}}}$, such that $|\nu_0^{\tilde{\mathcal{P}}}(\mathcal{P}) - \nu_0^{\tilde{\mathcal{P}}}(\mathcal{P}_{\mathcal{X}})| \leq \frac{C_0^{\tilde{\mathcal{P}}}}{\sqrt{K}}$ and $|\nu_1^{\tilde{\mathcal{P}}}(\mathcal{P}) - \nu_1^{\tilde{\mathcal{P}}}(\mathcal{P}_{\mathcal{X}})| \leq \frac{C_1^{\tilde{\mathcal{P}}}}{\sqrt{K}}$.*

---

**Theorem 3.2.** *Let Assumptions 1 and 3 hold. For **data generation process 2**, with probability at least $1 - \delta$,*

$$\mathcal{E}_d(f_{S^d} \circ h_{S_p}, \tilde{\mathcal{P}}) \leq \left( \nu_0^{\tilde{\mathcal{P}}}(\mathcal{P}_{\mathcal{X}}) + \frac{C_0^{\tilde{\mathcal{P}}}}{\sqrt{K}} \right) \left( 5M_\ell \sqrt{\frac{\log \frac{4}{\delta}}{2N}} + \frac{2G\sqrt{2}}{\sqrt{N}} \right) + \nu_1^{\tilde{\mathcal{P}}}(\mathcal{P}_{\mathcal{X}}) + \frac{C_1^{\tilde{\mathcal{P}}}}{\sqrt{K}}$$

$$+ 5M_\ell \sqrt{\frac{\log \frac{4}{\delta}}{2\tilde{N}}} + 2\sqrt{2}G\frac{1}{\sqrt{\tilde{N}}}.$$

---

According to Theorem 3.2, we can see that there is no longer a trade-off between the inter-class diversity $K$ and the intra-class diversity $n$. Instead, the bound gets smaller when $K$ is larger. To verify this, we conduct experiments under this setting (Figure 4 in the Appendix A.3), and observe a tendency of lower error rate with increasing $K$.

**Discussion.** The classes and data samples correspond to two dimensions of diversity. When the size of the pre-training dataset is fixed, in the case of Theorem 3.1, both diversities would vary in a see-saw-like way with different $K$, but for Theorem 3.2, since we fix the data samples used, only one dimension of diversity varies when $K$ is different and therefore the downstream performance only depends on the varying diversity ($K$). This difference reveals the underlying cause of the downstream performance trade-off rather than relying solely on $K$ and $n$ values.

## 4 Justification and Application

In this section, we first verify our theoretical findings via empirical results. Then, we show the effectiveness of using the optimal class-to-sample ratio, which is estimated based on our theory with a relatively small number of data samples, to build larger pre-training datasets.

## 4.1 Are the trade-off curves for different $N$ aligned?

According to our findings in Section 3.3, the test error on a downstream task is a convex function with respect to the class-to-sample ratio (Equation 4) and the optimal class-to-sample ratio $\bar{x} = \frac{B^2}{A^2}$ is invariant to the size of pre-training dataset $N$. To empirically verify this, we visualize the performance on downstream tasks as a function of the class-to-sample ratio with different $N$ in Figure 2.

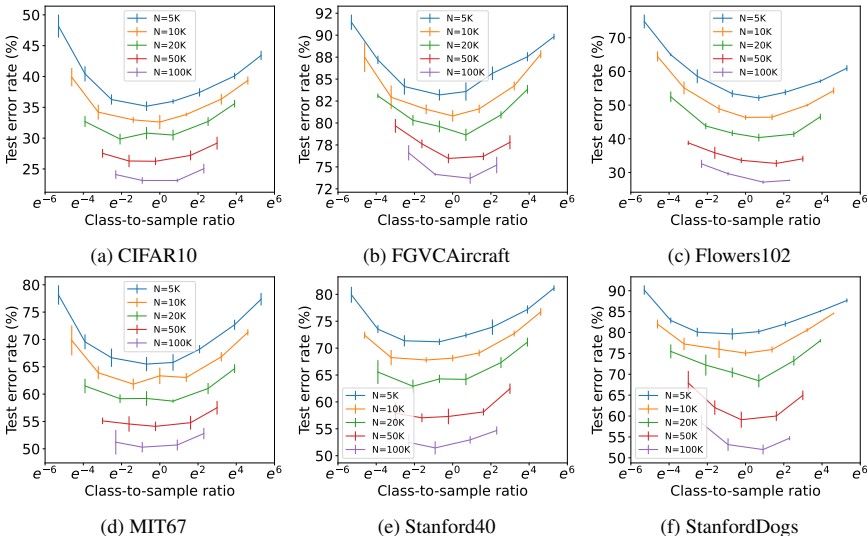

Figure 2: Test error rate across class-to-sample ratio. The vertical bar is the standard deviation.

From the figures, we can see that the curves of different $N$ for a specific downstream task are aligned, as well as the optimal class-to-ratios, which follows our theoretical findings. This indicates that empirically, one could extrapolate the optimal class-to-ratio estimated with a small $N$ for building a large-scale pre-training dataset. Note that the rightmost point of each curve corresponds to using all the classes in ImageNet, *i.e.*, $K = 1000$, and we can see that such a standard design choice does not lead to optimal downstream performance, especially with small pre-training datasets. In addition, we conducted experiments with different pre-trained datasets (Appendix A.4), different model backbones (Appendix A.6), and downstream tasks of different domains (Appendix A.5) to demonstrate that our findings are consistent across many scenarios.

## 4.2 Predicting the optimal number of pre-training classes

As a direct application of our theoretical and empirical findings, one could estimate the optimal class-to-sample ratio with a small $N$ and use it to decide the optimal number of classes when building a larger pre-training dataset. In particular, denoted by $\bar{x}$ the optimal class-to-sample ratio, the optimal number of classes for a given $N$ is $\bar{K} = \sqrt{\bar{x}N}$. We refer to this approach as **Extrapolation**. We empirically compare it against the following methods of deciding the number of classes when building a pre-training dataset: 1) **Standard:** the number of classes equals 1000 as the standard design choice of ImageNet; 2) **Grid Search:** the number of classes corresponding to the data point with the lowest error rate for each curve in Figure 2; 3) **Fitting:** given the target size, we use the corresponding data points in Figure 2 to fit the theoretically-derived performance function (Equation 4), and then analytically calculate the optimal number of classes. We use the calculated number of classes to build a pre-training dataset and measure the performance of a model trained with it. In reality, the latter two baselines require repeatedly training models on pre-training datasets with the target size yet different numbers of classes and are therefore time-consuming and data-intensive.

We set the target size of pre-training dataset as {50K, 100K} and round the number of classes to an integer if needed. For our Extrapolation method, we only use three data points with $N$ being much smaller than the target size to estimate the optimal class-to-sample ratio: $N = 5000$ and $K = \{10, 50, 200\}$. We use the estimated optimal class-to-sample ratio for both target sizes. The results as well as the number of classes selected by the above methods can be found in Table 1. From

Table 1: Test error rate on downstream tasks (the first row of each task, lower is better), and the number of classes in the pre-training dataset (the second row).

| $N$ | Method | Target Dataset | | | | | |
|---|---|---|---|---|---|---|---|
| | | CIFAR10 | FGVCAircraft | Flowers102 | MIT67 | Stanford40 | StanfordDogs |
| 50K | Standard ($K$=1000) | $29.19_{\pm0.14}$ | $77.80_{\pm0.28}$ | $34.08_{\pm0.35}$ | $57.51_{\pm0.18}$ | $62.45_{\pm0.36}$ | $64.96_{\pm0.38}$ |
| | Grid Search | $26.24_{\pm0.44}$ | $75.96_{\pm0.43}$ | $32.70_{\pm0.21}$ | $54.10_{\pm0.4}$ | $57.05_{\pm0.44}$ | $59.12_{\pm0.2}$ |
| | | (200) | (200) | (500) | (200) | (100) | (200) |
| | Fitting | $26.25_{\pm0.47}$ | $76.00_{\pm0.44}$ | $32.13_{\pm0.10}$ | $53.60_{\pm0.39}$ | $57.10_{\pm0.3}$ | $59.76_{\pm0.26}$ |
| | | (169) | (293) | (415) | (161) | (138) | (260) |
| | Extrapolation | $26.27_{\pm0.21}$ | $76.18_{\pm0.14}$ | $32.60_{\pm0.06}$ | $53.01_{\pm0.23}$ | $57.25_{\pm0.14}$ | $60.15_{\pm0.27}$ |
| | | (190) | (168) | (296) | (163) | (134) | (158) |
| 100K | Standard ($K$=1000) | $25.04_{\pm0.26}$ | $75.21_{\pm0.48}$ | $27.69_{\pm0.45}$ | $52.79_{\pm0.36}$ | $54.69_{\pm0.39}$ | $54.70_{\pm0.45}$ |
| | Grid Search | $23.13_{\pm0.06}$ | $73.70_{\pm0.50}$ | $27.15_{\pm0.44}$ | $50.30_{\pm0.46}$ | $51.45_{\pm0.33}$ | $51.98_{\pm0.30}$ |
| | | (500) | (500) | (500) | (200) | (200) | (500) |
| | Fitting | $22.67_{\pm0.33}$ | $73.45_{\pm0.33}$ | $26.67_{\pm0.33}$ | $50.82_{\pm0.33}$ | $52.24_{\pm0.33}$ | $52.24_{\pm0.33}$ |
| | | (276) | (372) | (655) | (249) | (207) | (392) |
| | Extrapolation | $23.12_{\pm0.13}$ | $73.30_{\pm0.17}$ | $26.98_{\pm0.41}$ | $50.42_{\pm0.25}$ | $52.32_{\pm0.06}$ | $53.17_{\pm0.26}$ |
| | | (269) | (238) | (418) | (231) | (190) | (233) |

Table 2: Total number of samples used for building the pre-training dataset.

| $N$ | Method | Target Dataset | | | | | |
|---|---|---|---|---|---|---|---|
| | | CIFAR10 | FGVCAircraft | Flowers102 | MIT67 | Stanford40 | StanfordDogs |
| 50K | Standard ($K$=1000) | 50K | | | | | |
| | Grid Search | 150K (5 trials) | | | | | |
| | Fitting | 158.164K | 161.570K | 159.403K | 158.694K | 159.268K | 160.538K |
| | Extrapolation | 55.418K | 55.183K | 55.830K | 55.117K | 54.598K | 55.045K |
| 100K | Standard ($K$=1000) | 100K | | | | | |
| | Grid Search | 260K (4 trials) | | | | | |
| | Fitting | 272.336K | 271.836K | 268.164K | 269.878K | 261.981K | 270.579K |
| | Extrapolation | 103.937K | 103.608K | 104.517K | 103.515K | 103.049K | 103.401K |

the results, we can see that although the ImageNet dataset is widely used, its number of classes ($K = 1000$) is not optimal for building a pre-training dataset of 50K/100K samples, since the Standard underperforms other methods. Besides, methods except for the Standard all render similar test error rate even though their number of classes are different, which reveals that the performance is not sensitive to the number of classes as long as we pick a reasonable number. Thus, our Extrapolation method is superior to Grid Search and Fitting, since it needs much fewer samples to estimate the number of classes, while both Grid Search and Fitting require building the pre-training dataset of target size multiple times.

## 4.3 Extra data needed for estimating the optimal class-to-sample ratio

One advantage of the Standard method is that it does not require extra data for estimating the optimal number of classes. In contrast, other methods would introduce extra data unused in the final pre-training dataset. For example, when estimating the optimal class-to-sample ratio, one may sample data from 1000 classes but eventually find the optimal number of classes is 600, then the data of the additional 400 classes would not be used in the final pre-training dataset. We then investigate how much data is needed by different methods to build the final pre-training dataset. We list the total number of samples used by different methods in Table 2. From the table, we can see that Grid Search and Fitting require much more samples than the target size of the pre-training dataset, since they involve building the pre-training dataset of the target size multiple times, while the number of extra data needed by Extrapolation is relatively small, because it estimates the optimal number of classes using a small $N$ of 5000. In addition, using Extrapolation, one only needs to estimate the optimal class-to-sample ratio once and then use it for building pre-training datasets with different sizes without re-estimation.

As the Extrapolation requires more data than the Standard, a fairer comparison between them needs to ensure the total number of data used is similar, and the Standard would have a slightly larger pre-training dataset since it does not spend any data budget for estimation. In Figure 3, we compare

the Extrapolation to the Standard whose pre-training dataset size is slightly larger than the total number of data used by the Extrapolation. Each bar plot represents a specific downstream task, and the number in parentheses indicates the size ($N$) of the corresponding pre-training dataset. We can observe that in most cases, the Extrapolation demonstrates improved performance over the Standard even when the latter uses a larger pre-training dataset, which further justifies the effectiveness of the Extrapolation method.

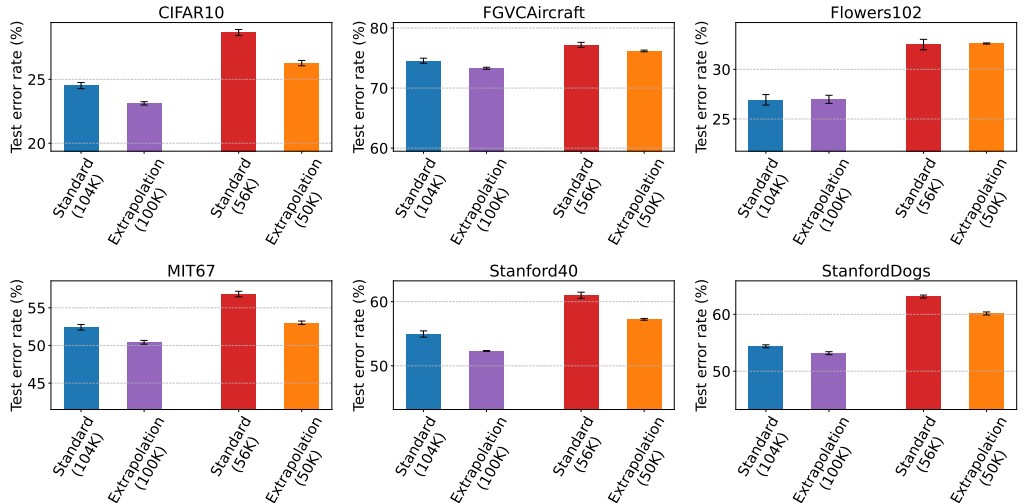

Figure 3: Each bar plot visualizes the error rates of different methods on a specific downstream task. The number in parentheses is $N$, i.e. the size of the corresponding pre-training dataset.

## 5  Related Work

We briefly review recent studies on supervised pre-training from data-centric perspectives. First, on the composition of the pre-training dataset, [9] presents a scaling law that predicts the test loss on downstream tasks under varying source dataset compositions, while [14] studies the performance on downstream tasks when subsets of the pre-training dataset are removed. Second, on the label space of supervised pre-training, [26] offers a statistical analysis explaining pre-training techniques' success in NLP, showing that class diversity in pre-training tasks substantially enhances sample efficiency in downstream tasks, while the study by [12] explores the impact of pre-training label granularity on downstream tasks, emphasizing the importance of selecting an appropriate level of label granularity. Lastly, [7] explores the impact of pre-training data distribution on transfer performance, finding the choice of the pre-training dataset to be crucial. In contrast, we dive into the trade-off of the intra-/inter-class diversity in the supervised pre-training dataset. The study related the most to ours is [13], where the authors empirically examined the importance of pre-training data characteristics on downstream performance. While covering a wide range of pre-training data characteristics, this study only briefly explores the trade-off of intra-/inter-class diversity in the pre-training dataset (Section 5.5 in [13]). Specifically, the authors only considered two different cases of intra-/inter-class diversity, *i.e.*, $K = \{500, 1000\}$ for ImageNet. In contrast, we, both empirically and theoretically, show how such a trade-off would impact the downstream performance and our theory uncovers a surprising property of the optimal class-to-sample ratio: it is invariant to the size of the pre-training dataset.

## 6  Conclusion

In this study, we explore the trade-off of the intra-/inter-class diversity in supervised pre-training datasets of fixed size. We discovered that the optimal downstream performance is achieved through a balance of intra-/inter-class diversity. Our theory demonstrates that downstream performance depends on both diversities, and the optimal class-to-sample ratio remains constant regardless of the dataset size. We apply this finding to predict the optimal number of classes in pre-training datasets and provide evidence of its effectiveness across many scenarios.

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

# A Appendix

## A.1 Limitation and Potential Negative Social Impact

Some potential negative societal impacts might arise from this research, such as:

**Bias Amplification**: When the research discusses the optimal class-to-sample ratio in pre-training datasets, it does not discuss how these classes are determined. There's a risk that the choice of classes and the samples within these classes could reflect and perpetuate existing biases in society. For instance, if the classes are determined by stereotypical or biased criteria, models trained on these datasets could amplify these biases in their predictions or recommendations.

**Overemphasis on Quantity over Quality**: This research might also create an overemphasis on the quantity (size and diversity) of the data at the expense of its quality. Poor data quality could lead to the development of inaccurate or unreliable machine learning models.

## A.2 Experimental Details

### A.2.1 Training Details

We build our code on Python and Pytorch. We fix the model to be the ResNet-18 [10]. For pre-training, we set the number of epochs to be 100 and the batch size to be 64. We use the Adam optimizer for training with a learning rate of 0.1, a momentum of 0.9, and a weight decay of 1e-4. We repeat each experiment 3 times with different seeds and report the mean and variance of the results. All experiments ran on a machine with an Intel(R) Xeon(R) CPU E5-2678 v3 with 512G memory and two 48G NVIDIA RTX A6000 GPUs.

### A.2.2 Details of Dataset

**The Pre-training Dataset**:

- **ImageNet** [4]. It is an image dataset organized according to the WordNet hierarchy. Each meaningful concept in WordNet, possibly described by multiple words or word phrases, is called a "synonym set" or "synset". There are more than 100,000 synsets in WordNet; the majority of them are nouns (80,000+).
- **Place365** [25]. It has 1,803,460 training images with the image number per class varying from 3,068 to 5,000. The validation set has 50 images per class and the test set has 900 images per class.

**The Downstream Tasks Dataset**

- **Stanford Actions 40** [25]. It contains images of humans performing 40 actions. There are about 180-300 images per class. We do not use bounding boxes and other annotation information for training. There are a total of 9,532 images, making it the smallest dataset in our benchmark experiments.
- **Stanford Dogs 120** [15]. It contains images of 120 breeds of dogs worldwide. There are precisely 100 examples per category in the training set. It is used for the task of fine-grained image categorization. We do not use the bounding box annotations. There are a total of 20,580 images.
- **MIT Indoors 67** [22]. It is a scene classification dataset containing 67 indoor scene categories, each consisting of 80 images for training and 20 for testing. Indoor scene recognition is challenging because spatial properties, background information, and object characters are expected to be extracted. There are 15,620 images in total.
- **CIFAR10** [16]. It is a collection of images commonly used to train machine learning and computer vision algorithms. It contains 60,000 32x32 color images in 10 different classes. The ten classes represent airplanes, cars, birds, cats, deer, dogs, frogs, horses, ships, and trucks. There are 6,000 images of each class.
- **Flowers102** [20]. It is an image classification dataset consisting of 102 flower categories. The flowers are chosen to be flowers commonly occurring in UK. Each class consists of

between 40 and 258 images. The images have large scale, pose and light variations. In addition, some categories have significant variations within the category and several very similar categories.

- **FGVCAircraft** [18]. It contains 10,200 images of aircraft, with 100 images for each of 102 different aircraft model variants, most of which are airplanes. Each image's (main) aircraft is annotated with a tight bounding box and a hierarchical airplane model label. Aircraft models are organized in a four-level hierarchy.

- **DomainNet-real and DomainNet-painting** [21]. DomainNet-real and DomainNet-painting belong to real and painting domains, respectively, and both comprise 345 categories. DomainNet-real contains over 170k images, while DomainNet-painting has more than 70k images.

### A.3 Empirical Verification of Theorem 3.2

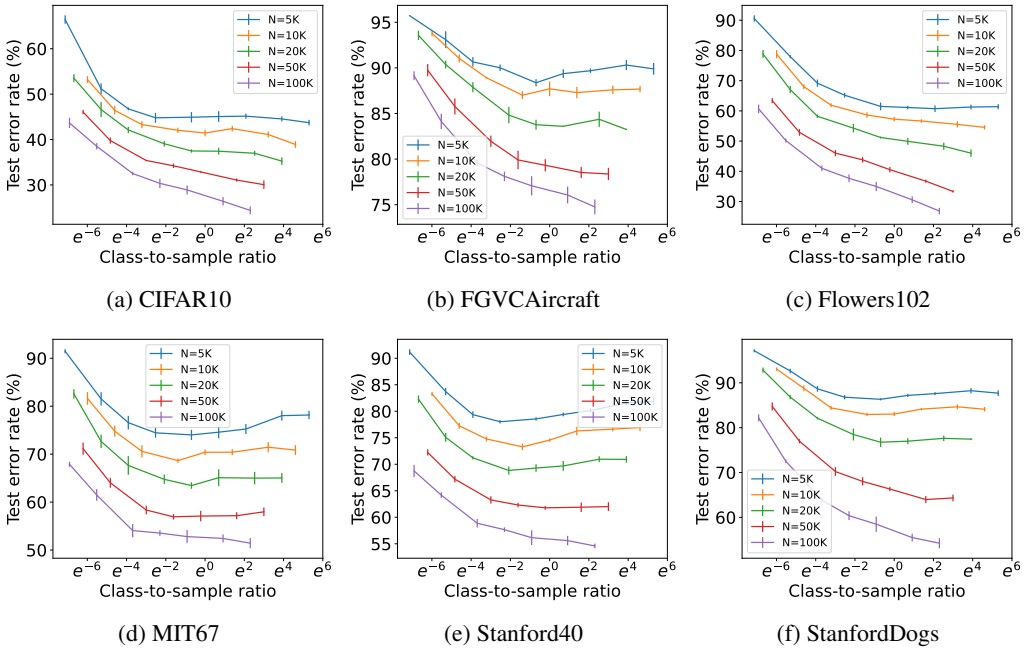

Figure 4: Test error rate across class-to-sample ratio. The vertical bar at each point is the standard deviation.

As shown in Figure 4, we can find that, across all datasets, the results generally showed a decreasing trend in test error rate as the class-to-sample ratio increased. This empirically supports the theoretical assertion that there is no downstream performance trade-off caused by the inter-class diversity $K$ and the intra-class diversity $n$. Instead, the downstream performance improves with increasing $K$, as suggested in Theorem 3.2.

### A.4 Experiments on Places365 as Pre-training Dataset

We use the Places365[3] [27] for our pre-training and the result is represented in Figure 5. We then performed evaluations on the same batch of downstream tasks as in the main body of the paper to demonstrate the trade-off between intra- and inter-class diversity is consistent with our main findings.

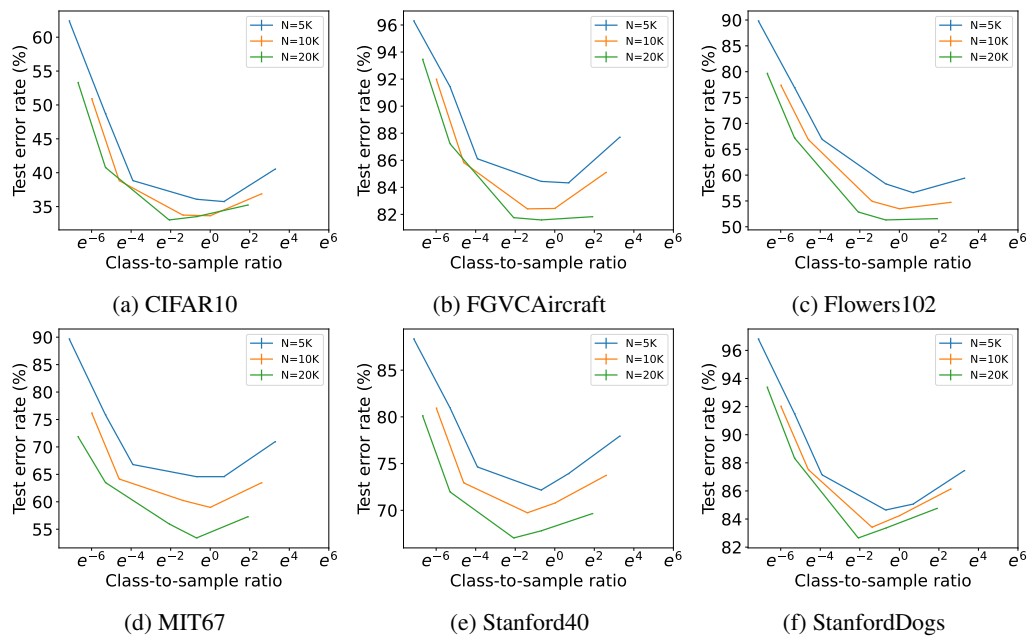

Figure 5: Test error rate across class-to-sample ratio. ResNet-18 pre-trained on the Places365 dataset.

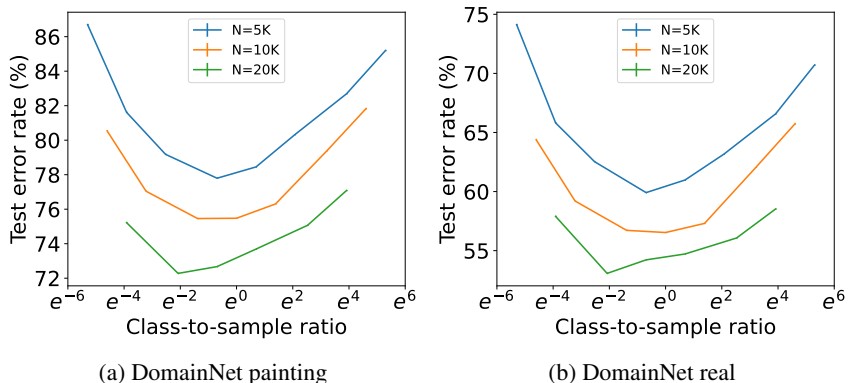

Figure 6: Test error rate across class-to-sample ratio. ResNet-18 pre-trained on ImageNet and tested on two datasets of different domains from the DomainNet benchmark.

## A.5   Experiments on Downstream Tasks of Different Domains

As illustrated in Figure 6, we test models trained on ImageNet on two distinct domains (painting and real images) sourced from the DomainNet benchmark[4] [21] and the result indicates that our conclusions do not change with different downstream task datasets.

## A.6   Experiments on ViT as Model Backbone

To ensure our conclusions are not restricted to specific model architecture, we use ViT-B-16 model [5] as the backbone model. The results in Figure 7 show that different model backbone does not change the conclusions we derived.

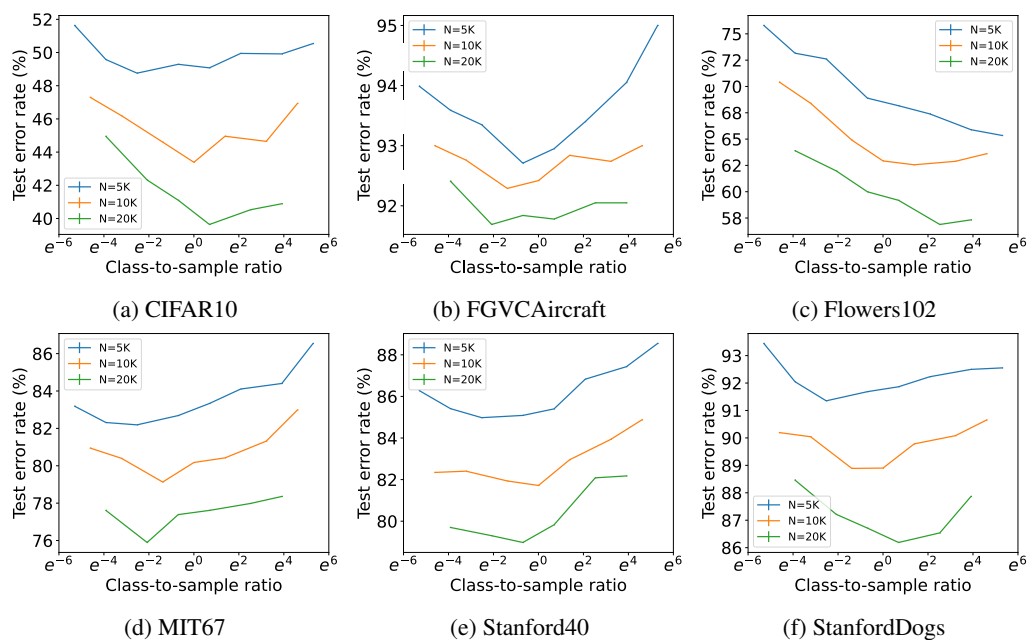

Figure 7: Test error rate across class-to-sample ratio. ViT-B-16 pre-trained on the ImageNet dataset.

Table 3: Actual v.s. predicted test error rate on downstream tasks according to Equation 4.

| $N$ | Method | Target Dataset | | | | | |
|---|---|---|---|---|---|---|---|
| | | CIFAR10 | FGVCAircraft | Flowers102 | MIT67 | Stanford40 | StanfordDogs |
| 50K | Predicted | $26.13_{\pm 0.1}$ | $75.95_{\pm 0.4}$ | $32.96_{\pm 0.48}$ | $53.99_{\pm 0.43}$ | $56.72_{\pm 0.16}$ | $58.87_{\pm 0.35}$ |
| | Actual | $26.25_{\pm 0.47}$ | $76.00_{\pm 0.44}$ | $32.13_{\pm 0.10}$ | $53.60_{\pm 0.39}$ | $57.10_{\pm 0.3}$ | $59.76_{\pm 0.26}$ |
| 100K | Predicted | $22.84_{\pm 0.40}$ | $73.52_{\pm 0.41}$ | $27.33_{\pm 0.40}$ | $50.15_{\pm 0.43}$ | $51.74_{\pm 0.36}$ | $51.69_{\pm 0.23}$ |
| | Actual | $22.67_{\pm 0.33}$ | $73.45_{\pm 0.33}$ | $26.67_{\pm 0.33}$ | $50.82_{\pm 0.33}$ | $52.24_{\pm 0.33}$ | $52.24_{\pm 0.33}$ |

## A.7 Actual v.s. Predicted Performance of Fitting Equation 4

In Section 4.2, we study the performance of the Fitting method, *i.e.*, using the optimal number of classes derived from the function fitting Equation 4. Given the fitted function, one can also predict the test error given a specific value of the number of classes. Here, we compare the actual test error of the derived number of classes (the Fitting entry of Table 1) with the one predicted by the fitted function, in order to see whether our theory offers an accurate prediction of performance. From the results presented in Table 3, we can see that the actual test error (Actual) is similar to the test error predicted by the fitted function (Predicted), this further proves the utility of our theory.

## A.8 Proofs of Theoretical Results

### A.8.1 Proof of Theorem 3.1

Before we formally state the proof of Theorem 3.1, we define several notations as follows:

$$h_{\mathcal{P}} \triangleq \arg \min_{h \in \mathcal{H}} \mathcal{R}_p(h, \mathcal{P}),$$

$$f_{\mathcal{P}} \triangleq \arg \min_{f} \mathbb{E}_{i \sim \mathrm{Unif}[K], x \sim \mathcal{P}_i}[\ell(f \circ h_{\mathcal{P}}(x), i)].$$

*Proof of Theorem 3.1.* **Step I. Bound $\mathcal{E}_p(h_{S^p}, \mathcal{P})$.**

---

[3]http://places2.csail.mit.edu/

[4]http://ai.bu.edu/M3SDA/

$\mathcal{E}_p(h_{S^p}, \mathcal{P})$ can be decomposed into

$$\mathcal{E}_p(h_{S^p}, \mathcal{P}) = \mathcal{R}_p(h_{S^p}, \mathcal{P}) - \min_{\tilde{h} \in \mathcal{H}} \mathcal{R}_p(\tilde{h}, \mathcal{P})$$

$$= \left( \mathcal{R}_p(h_{S^p}, \mathcal{P}) - \frac{1}{nK} \sum_{i=1}^{K} \sum_{j=1}^{n} \ell(f_{S^p} \circ h_{S^p}(x_{i,j}), i) \right)$$

$$+ \left( \frac{1}{nK} \sum_{i=1}^{K} \sum_{j=1}^{n} \ell(f_{S^p} \circ h_{S^p}(x_{i,j}), i) - \frac{1}{nK} \sum_{i=1}^{K} \sum_{j=1}^{n} (\ell(f_{\mathcal{P}} \circ h_{\mathcal{P}}(x_{i,j}), i)) \right)$$

$$+ \left( \frac{1}{nK} \sum_{i=1}^{K} \sum_{j=1}^{n} (\ell(f_{\mathcal{P}} \circ h_{\mathcal{P}}(x_{i,j}), i)) - \mathcal{R}_p(h_{\mathcal{P}}, \mathcal{P}) \right).$$

We tackle the three terms of the RHS of the above inequality respectively. As for the first term, since

$$\mathcal{R}_p(h_{S^p}, \mathcal{P}) = \min_{f \in \mathcal{F}} \mathbb{E}_{i \sim \mathrm{Unif}[K], x \sim \mathcal{P}_i}[\ell((f \circ h_{S^p}(x), i)]$$

$$\leq \mathbb{E}_{i \sim \mathrm{Unif}[K], x \sim \mathcal{P}_i}[\ell(f_{S^p} \circ h_{S^p}(x), i)],$$

we have that the first term can be bounded by

$$\mathcal{R}_p(h_{S^p}, \mathcal{P}) - \frac{1}{nK} \sum_{i=1}^{K} \sum_{j=1}^{n} \ell(f_{S^p} \circ h_{S^p}(x_{i,j}), i)$$

$$\leq \mathbb{E}_{i \sim \mathrm{Unif}[K], x \sim \mathcal{P}_i}[\ell(f_{S^p} \circ h_{S^p}(x), i)] - \frac{1}{nK} \sum_{i=1}^{K} \sum_{j=1}^{n} \ell(f_{S^p} \circ h_{S^p}(x_{i,j}), i)$$

$$\leq \sup_{f \in \mathcal{F}, h \in \mathcal{H}} \left[ \mathbb{E}_{i \sim \mathrm{Unif}[K], x \sim \mathcal{P}_i}[\ell((f \circ h(x), i)] - \frac{1}{nK} \sum_{i=1}^{K} \sum_{j=1}^{n} \ell(f \circ h(x_{i,j}), i) \right].$$

Applying Gaussian complexity to $\frac{1}{n} \sum_{j=1}^{n} (\frac{1}{K} \sum_{i=1}^{K} \ell(f \circ h(x_{i,j}), i))$, we obtain that with probability at least $1 - \delta$, the RHS of the above inequality is smaller than

$$M_\ell \sqrt{\frac{9 \log \frac{1}{\delta}}{2n}} + 2\mathbb{E}_{((x_{i,j})_{i=1}^{K})_{j=1}^{n} \sim (\Pi_{i=1}^{K} \mathcal{P}_i^n)} \mathbb{E}_{(\sigma_i)_{i=1}^{n} \sim \mathcal{N}(0, \mathbb{1}_{n \times n})} \sup_{f \in \mathcal{F}, h \in \mathcal{H}} \frac{1}{n} \sum_{j=1}^{n} \sigma_j \left( \frac{1}{K} \sum_{i=1}^{K} \ell(f \circ h(x_{i,j}), i) \right)$$

$$\leq M_\ell \sqrt{\frac{9 \log \frac{1}{\delta}}{2n}} + \frac{2}{K} \sum_{i=1}^{K} \mathbb{E}_{(x_{i,j})_{j=1}^{n} \sim \mathcal{P}_i^n} \mathbb{E}_{(\sigma_i)_{i=1}^{n} \sim \mathcal{N}(0, \mathbb{1}_{n \times n})} \sup_{f \in \mathcal{F}, h \in \mathcal{H}} \frac{1}{n} \sum_{j=1}^{n} \sigma_j \ell(f \circ h(x_{i,j}), i)$$

$$\leq M_\ell \sqrt{\frac{9 \log \frac{1}{\delta}}{2n}} + \frac{2\sqrt{2}}{K} \sum_{i=1}^{K} \mathbb{E}_{(x_{i,j})_{j=1}^{n} \sim \mathcal{P}_i^n} \mathbb{E}_{(\sigma_{j,l})_{j \in [n], l \in [K]} \sim \mathcal{N}(0, \mathbb{1}_{nK})} \sup_{f \in \mathcal{F}, h \in \mathcal{H}} \frac{1}{n} \sum_{j=1}^{n} \sum_{l=1}^{K} \sigma_{j,l} f_l \circ h(x_{i,j})$$

$$\leq M_\ell \sqrt{\frac{9 \log \frac{1}{\delta}}{2n}} + 2G\sqrt{2}\frac{1}{\sqrt{n}}.$$

Here the second inequality is due to Slepian's Lemma, and the last inequality is due to Assumption 1. All in all, with probability at least $1 - \delta$, the first term can be bounded as

$$\mathcal{R}_p(h_{S^p}, \mathcal{P}) - \frac{1}{nK} \sum_{i=1}^{K} \sum_{j=1}^{n} \ell(f_{S^p} \circ h_{S^p}(x_{i,j}), i) \leq M_\ell \sqrt{\frac{9 \log \frac{1}{\delta}}{2n}} + 2G\sqrt{2}\frac{1}{\sqrt{n}}.$$

Meanwhile, the second term is non-positive due to the optimality of $f_{S^p}$ and $h_{S^p}$.

Finally, the third term can be bounded as

$$\left( \frac{1}{nK} \sum_{i=1}^{K} \sum_{j=1}^{n} \left( \ell(f_{\mathcal{P}} \circ h_{\mathcal{P}}(x_{i,j}), i) \right) - \mathcal{R}_p(h_{\mathcal{P}}, \mathcal{P}) \right)$$

$$= \left( \frac{1}{nK} \sum_{i=1}^{K} \sum_{j=1}^{n} \left( \ell(f_{\mathcal{P}} \circ h_{\mathcal{P}}(x_{i,j}), i) \right) - \mathbb{E}_{i \sim \mathrm{Unif}[K], x \sim \mathcal{P}_i}[\ell(f_{\mathcal{P}} \circ h_{\mathcal{P}}(x), i)] \right)$$

$$\overset{w.p.1-\delta}{\leq} M_{\ell} \sqrt{\frac{2 \log \frac{1}{\delta}}{n}},$$

where the last inequality is due to Hoeffding's inequality. As a conclusion of Stage I, we obtain that with probability at least $1 - 2\delta$,

$$\mathcal{E}_p(h_{S^p}, \mathcal{P}) \leq 5 M_{\ell} \sqrt{\frac{\log \frac{1}{\delta}}{2n}} + \frac{2G\sqrt{2}}{\sqrt{n}}.$$

**Step II. Bound $\mathcal{E}_d(h_{S^p})$.** Applying Assumption 2, we obtain that with probability at least $1 - 2\delta$,

$$\mathcal{E}_d(h_{S^p}, \tilde{\mathcal{P}}) \leq \nu_1^{\tilde{\mathcal{P}}}(\mathcal{P}) \mathcal{E}_p(h_{S^p}, \mathcal{P}_{i_1}, \mathcal{P}_{i_2}) + \nu_0^{\tilde{\mathcal{P}}}(\mathcal{P}) \leq \nu_1^{\tilde{\mathcal{P}}}(\mathcal{P}) \left( 5 M_{\ell} \sqrt{\frac{\log \frac{1}{\delta}}{2n}} + \frac{2G\sqrt{2}}{\sqrt{n}} \right) + \nu_0^{\tilde{\mathcal{P}}}(\mathcal{P}).$$

By McDiarmid's inequality, we obtain that with probability at least $1 - \delta$,

$$\nu_0^{\tilde{\mathcal{P}}}(\mathcal{P}) \leq \mathbb{E}_{\mathcal{P} \sim \mathcal{D}^K} \nu_0^{\tilde{\mathcal{P}}}(\mathcal{P}) + M_0 \sqrt{\frac{\log \frac{1}{\delta}}{2K}} \leq \nu_0^{\tilde{\mathcal{P}}}(\mathcal{D}) + M_0 \sqrt{\frac{\log \frac{1}{\delta}}{2K}} + \frac{C_0}{\sqrt{K}}.$$

Similarly, with probability at least $1 - \delta$,

$$\nu_1^{\tilde{\mathcal{P}}}(\mathcal{P}) \leq \mathbb{E}_{\mathcal{P} \sim \mathcal{D}^K} \nu_1^{\tilde{\mathcal{P}}}(\mathcal{P}) + M_1 \sqrt{\frac{\log \frac{1}{\delta}}{2K}} \leq \nu_1^{\tilde{\mathcal{P}}}(\mathcal{D}) + M_1 \sqrt{\frac{\log \frac{1}{\delta}}{2K}} + \frac{C_1}{\sqrt{K}}.$$

As a conclusion, we obtain that with probability at least $1 - 4\delta$,

$$\mathcal{E}_d(h_{S^p}, \tilde{\mathcal{P}})$$

$$\leq \left( \nu_1^{\tilde{\mathcal{P}}}(\mathcal{D}) + M_1 \sqrt{\frac{\log \frac{1}{\delta}}{2K}} + \frac{C_1}{\sqrt{K}} \right) \left( 5 M_{\ell} \sqrt{\frac{\log \frac{1}{\delta}}{2n}} + \frac{2G\sqrt{2}}{\sqrt{n}} \right) + \nu_0^{\tilde{\mathcal{P}}}(\mathcal{D}) + M_0 \sqrt{\frac{\log \frac{1}{\delta}}{2K}} + \frac{C_0}{\sqrt{K}}.$$

**Step III. Bound $\bar{\mathcal{R}}_d(f_{S^d} \circ h_{S_p}, S_p) - \mathcal{R}_d(h_{S_p}, \tilde{\mathcal{P}})$** Finally, denote $f_{\tilde{\mathcal{P}}} \triangleq \arg\min_{\tilde{f}} \min_{\tilde{f} \in \tilde{\mathcal{F}}} \mathcal{R}_d(\tilde{f} \circ h, \tilde{\mathcal{P}})$ based on Hoeffding's inequality, we obtain that with probability at least $1 - \delta$,

$$\bar{\mathcal{R}}_d(f_{S^d} \circ h_{S_p}, S_p) - \mathcal{R}_d(h_{S_p}, \tilde{\mathcal{P}}) \leq \mathcal{R}_d(f_{\tilde{\mathcal{P}}} \circ h_{S_p}, \tilde{\mathcal{P}}) - \mathcal{R}_d(h_{S_p}, \tilde{\mathcal{P}})$$

$$\leq M_{\ell} \sqrt{\frac{2 \log \frac{1}{\delta}}{\tilde{N}}}.$$

Meanwhile, applying Gaussian's inequality, we have that with probability at least $1 - \delta$,

$$\mathcal{R}_d(f_{S^d} \circ h_{S_p}, \tilde{\mathcal{P}}) - \bar{\mathcal{R}}_d(f_{S^d} \circ h_{S_p}, S_p)$$

$$\leq 2 \mathbb{E}_{(x_i, y_i)_{i=1}^{\tilde{N}} \sim \tilde{\mathcal{P}}^{\tilde{N}}} \mathbb{E}_{(\sigma_i)_{i=1}^{\tilde{N}} \sim \mathcal{N}(0, \mathbb{1}_{\tilde{N} \times \tilde{N}})} \sup_{f \in \tilde{\mathcal{F}}, h \in \mathcal{H}} \frac{1}{\tilde{N}} \sum_{i=1}^{\tilde{N}} \sigma_i \ell(f \circ h(x_i), y_i) + M_{\ell} \sqrt{\frac{9 \log \frac{1}{\delta}}{2\tilde{N}}}$$

$$\leq 2\sqrt{2} \mathbb{E}_{(x_i, y_i)_{i=1}^{\tilde{N}} \sim \tilde{\mathcal{P}}^{\tilde{N}}} \mathbb{E}_{(\sigma_{i,j})_{i \in [\tilde{N}], j \in [\tilde{K}]} \sim \mathcal{N}(0, \mathbb{1}_{\tilde{N}\tilde{K} \times \tilde{N}\tilde{K}})} \sup_{f \in \tilde{\mathcal{F}}, h \in \mathcal{H}} \frac{1}{\tilde{N}} \sum_{i=1}^{\tilde{N}} \sum_{j=1}^{\tilde{K}} \sigma_{i,j} f_j \circ h(x_i) + M_{\ell} \sqrt{\frac{9 \log \frac{1}{\delta}}{2\tilde{N}}}$$

$$\leq 2G\sqrt{2} \frac{1}{\sqrt{\tilde{N}}} + M_{\ell} \sqrt{\frac{9 \log \frac{1}{\delta}}{2\tilde{N}}},$$

where the second inequality is due to Slepian's Lemma, and the last inequality is due to Assumption 1.

All in all, we have with probability at least $1 - 6\delta$,

$$
\begin{aligned}
&\mathcal{E}_d(f_{S^d} \circ h_{S_p}, \tilde{\mathcal{P}}) \\
=&\mathcal{E}_d(h_{S^p}, \tilde{\mathcal{P}}) + \mathcal{R}_d(f_{S^d} \circ h_{S_p}, \tilde{\mathcal{P}}) - \bar{\mathcal{R}}_d(f_{S^d} \circ h_{S_p}, S_p) + \bar{\mathcal{R}}_d(f_{S^d} \circ h_{S_p}, S_p) - \mathcal{R}_d(h_{S_p}, \tilde{\mathcal{P}}) \\
\leq & \left( \nu_1^{\tilde{\mathcal{P}}}(\mathcal{D}) + M_1 \sqrt{\frac{\log\frac{1}{\delta}}{2K}} + \frac{C_1}{\sqrt{K}} \right) \left( 5M_\ell \sqrt{\frac{\log\frac{1}{\delta}}{2n}} + \frac{2G\sqrt{2}}{\sqrt{n}} \right) + \nu_0^{\tilde{\mathcal{P}}}(\mathcal{D}) + M_0 \sqrt{\frac{\log\frac{1}{\delta}}{2K}} + \frac{C_0}{\sqrt{K}} \\
& + 5M_\ell \sqrt{\frac{\log\frac{1}{\delta}}{2\tilde{N}}} + 2\sqrt{2}G \frac{1}{\sqrt{\tilde{N}}}.
\end{aligned}
$$

The proof is completed. $\qquad\square$

### A.8.2 Proof of Theorem 3.2

*Proof.* Denote

$$
h_{\mathcal{P}} \triangleq \arg\min_{h \in \mathcal{H}} \mathcal{R}_p(h),
$$

$$
f_{\mathcal{P}} \triangleq \arg\min_{f} \mathbb{E}_{(x,y)\sim\mathcal{P}}[\ell(f \circ h_{\mathcal{P}}(x), y)].
$$

**Step I. Bound $\mathcal{E}_p(h_{S^p})$.**

Denote $\mathcal{E}_p(h_{S^p})$ can be decomposed into

$$
\begin{aligned}
\mathcal{E}_p(h_{S^p}) =& \mathcal{R}_p(h_{S^p}) - \min_{\tilde{h}\in\mathcal{H}} \mathcal{R}_p(\tilde{h}) \\
=& \left( \mathcal{R}_p(h_{S^p}) - \frac{1}{N}\sum_{i=1}^{N} \ell(f_{S^p} \circ h_{S^p}(x_i), y_i) \right) \\
& + \left( \frac{1}{N}\sum_{i=1}^{N} \ell(f_{S^p} \circ h_{S^p}(x_i), y_i) - \frac{1}{N}\sum_{i=1}^{N} (\ell(f_{\mathcal{P}} \circ h_{\mathcal{P}}(x_i), y_i)) \right) \\
& + \left( \frac{1}{N}\sum_{i=1}^{N} (\ell(f_{\mathcal{P}} \circ h_{\mathcal{P}}(x_i), y_i)) - \mathcal{R}_p(h_{\mathcal{P}}) \right).
\end{aligned}
$$

We tackle the three terms of the RHS of the above inequality respectively. As for the first term, since

$$
\mathcal{R}_p(h_{S^p}) = \min_{f\in\mathcal{F}} \mathbb{E}_{(x,y)\sim\mathcal{P}}[\ell(f \circ h_{S^p}(x), y)] \leq \mathbb{E}_{(x,y)\sim\mathcal{P}}[\ell(f_{S^p} \circ h_{S^p}(x), y)],
$$

we have that the first term can be bounded by

$$
\begin{aligned}
& \mathcal{R}_p(h_{S^p}) - \frac{1}{N}\sum_{i=1}^{N} \ell(f_{S^p} \circ h_{S^p}(x_i), y_i) \\
\leq & \mathbb{E}_{(x,y)\sim\mathcal{P}}[\ell(f_{S^p} \circ h_{S^p}(x), i)] - \frac{1}{N}\sum_{i=1}^{N} \ell(f_{S^p} \circ h_{S^p}(x_i), y_i) \\
\leq & \sup_{f\in\mathcal{F}, h\in\mathcal{H}} \left[ \mathbb{E}_{(x,y)\sim\mathcal{P}}[\ell(f \circ h(x), i)] - \frac{1}{N}\sum_{i=1}^{N} \ell(f \circ h(x_i), y_i) \right].
\end{aligned}
$$

Applying Gaussian complexity to $\frac{1}{N}\sum_{i=1}^{N}\ell(f\circ h(x_i),y_i)$, we obtain that with probability at least $1-\delta$, the RHS of the above inequality is smaller than

$$M_\ell\sqrt{\frac{9\log\frac{1}{\delta}}{2N}}+2\mathbb{E}_{(x_i)_{i=1}^N\sim P^N}\mathbb{E}_{(\sigma_i)_{i=1}^N\sim\mathcal{N}(0,\mathbb{1}_N)}\sup_{f\in\mathcal{F},h\in\mathcal{H}}\frac{1}{N}\sum_{i=1}^{n}\sigma_i\ell(f\circ h(x_i),y_i)$$

$$\leq M_\ell\sqrt{\frac{9\log\frac{1}{\delta}}{2N}}+2\sqrt{2}\mathbb{E}_{(x_i)_{i=1}^N\sim P^N}\mathbb{E}_{(\sigma_{i,j})_{i\in[N],j\in[\dim(\mathcal{Y})]}\sim\mathcal{N}(0,\mathbb{1}_{N\dim(\mathcal{Y})})}\sup_{f\in\mathcal{F},h\in\mathcal{H}}\frac{1}{N}\sum_{i=1}^{n}\sum_{j=1}^{K}\sigma_{i,j}f_j\circ h(x_i)$$

$$\leq M_\ell\sqrt{\frac{9\log\frac{1}{\delta}}{2N}}+2G\sqrt{2}\frac{1}{\sqrt{N}}.$$

Here the first inequality is due to Slepian's lemma, and the last inequality is due to Assumption 1. All in all, with probability at least $1-\delta$, the first term can be bounded as

$$\mathcal{R}_p(h_{S^p})-\frac{1}{N}\sum_{i=1}^{N}\ell(f_{S^p}\circ h_{S^p}(x_i),y_i)\leq M_\ell\sqrt{\frac{9\log\frac{1}{\delta}}{2N}}+2G\sqrt{2}\frac{1}{\sqrt{N}}.$$

Meanwhile, the second term is non-positive due to the optimality of $f_{S^p}$ and $h_{S^p}$.

Finally, the third term can be bounded as

$$\frac{1}{N}\sum_{i=1}^{N}\ell(f_\mathcal{P}\circ h_\mathcal{P}(x_i),y_i)-\mathcal{R}_p(h_\mathcal{P})$$

$$=\frac{1}{N}\sum_{i=1}^{N}\ell(f_\mathcal{P}\circ h_\mathcal{P}(x_i),y_i)-\mathbb{E}_{(x,y)\sim\mathcal{P}}[\ell(f_\mathcal{P}\circ h_\mathcal{P}(x),y)]$$

$$\overset{w.p.1-\delta}{\leq}M_\ell\sqrt{\frac{2\log\frac{1}{\delta}}{N}},$$

where the last inequality is due to Hoeffding's inequality. As a conclusion of Stage I, we obtain that with probability at least $1-2\delta$,

$$\mathcal{E}_p(h_{S^p},\mathcal{P})\leq 5M_\ell\sqrt{\frac{\log\frac{1}{\delta}}{2N}}+\frac{2G\sqrt{2}}{\sqrt{N}}.$$

**Step II. Bound $\mathcal{E}_d(h_{S^p})$.** Applying Assumption 3, we obtain that with probability at least $1-2\delta$,

$$\mathcal{E}_d(h_{S^p},\tilde{\mathcal{P}})\leq\nu_1^{\tilde{\mathcal{P}}}(\mathcal{P})\mathcal{E}_p(h_{S^p})+\nu_0^{\tilde{\mathcal{P}}}(\mathcal{P})\leq\nu_1^{\tilde{\mathcal{P}}}(\mathcal{P})\left(5M_\ell\sqrt{\frac{\log\frac{1}{\delta}}{2N}}+\frac{2G\sqrt{2}}{\sqrt{N}}\right)+\nu_0^{\tilde{\mathcal{P}}}(\mathcal{P}).$$

Furthermore, as $|\nu_0^{\tilde{\mathcal{P}}}(\mathcal{P})-\nu_0^{\tilde{\mathcal{P}}}(\mathcal{P}_\mathcal{X})|\leq\frac{C_0^{\tilde{\mathcal{P}}}}{\sqrt{K}}$ and $|\nu_1^{\tilde{\mathcal{P}}}(\mathcal{P})-\nu_1^{\tilde{\mathcal{P}}}(\mathcal{P}_\mathcal{X})|\leq\frac{C_1^{\tilde{\mathcal{P}}}}{\sqrt{K}}$, we obtain that with probability at least $1-2\delta$

$$\mathcal{E}_d(h_{S^p},\tilde{\mathcal{P}})\leq\left(\nu_0^{\tilde{\mathcal{P}}}(\mathcal{P}_\mathcal{X})+\frac{C_0^{\tilde{\mathcal{P}}}}{\sqrt{K}}\right)\left(5M_\ell\sqrt{\frac{\log\frac{2}{\delta}}{2N}}+\frac{2G\sqrt{2}}{\sqrt{N}}\right)+\nu_1^{\tilde{\mathcal{P}}}(\mathcal{P}_\mathcal{X})+\frac{C_1^{\tilde{\mathcal{P}}}}{\sqrt{K}}.$$

The rest of the proof flows exactly the same as that of Theorem 3.1. $\square$

