# OpenReview forum: "On the Trade-off of Intra-/Inter-class Diversity for Supervised Pre-training"
_NeurIPS.cc/2023/Conference — NeurIPS 2023 poster_

### Official Review · Reviewer_AXB6 · 2023-07-05

**Soundness:** 3 good
**Presentation:** 3 good
**Contribution:** 3 good
**Rating:** 4
**Confidence:** 5

**Summary:**

This paper provides empirical study and theoretical analysis of the class-to-sample ratio in supervised pre-training datasets.

**Strengths:**

1. This papers reveals an important conclusion - the optimal class-to-sample ratio is invariant to the size of the pre-training dataset. This can be utilized as a guidance of the data size for the scaling law in pre-training.
2. Detailed theoretical analysis are provided, from which a guidance of choosing the class-to-sample ratio is also provided.

**Weaknesses:**

1. The pre-training dataset is restricted to ImageNet.
2. The model ResNet18 used is relatively small and may not reveal the true conclusion.
3. The evaluation is restricted to only 7 small downstream datasets.

**Questions:**

1. Would other datasets affect the conclusion drawn.
2. Would other larger models and vision transformers lead to the similar observation and conclusion?
3. Would similar observation holds on larger downstream datasets or downstream datasets of other domains?

**Limitations:**

See weakness and questions.

---

> ### Author Rebuttal · Authors · 2023-08-09
>
> In the general rebuttal, we added new experiments regarding the three weakness points/questions: using different pre-training dataset (**Figure R1**), using different model architecture (vision transformer, **Figure R2**), and using larger downstream datasets or datasets from other domain (painting domain, **Figure R3**). From the results, we observed that our main findings (the diversity trade-off and aligned optimal class-to-sample ratio) still hold for these different setups. Please find the details in the general rebuttal and we hope that your concern can be addressed by the new experiments.

---

### Official Review · Reviewer_Ypk9 · 2023-07-06

**Soundness:** 3 good
**Presentation:** 3 good
**Contribution:** 3 good
**Rating:** 6
**Confidence:** 4

**Summary:**

The paper presents a detailed analysis on the intra-class diversity and inter-class diversity when you have a fixed dataset size. The authors develop some theoretical rule based on empirical results.

**Strengths:**

- Provides a rigorous study on pretraining using ImageNet, that makes the empirical motivation clear for theoretical result.
- Theory is relatively well-explained in a step-by-step manner.
Proposes a method to balance intra-class and inter-class diversity, which leads to better performance on downstream tasks.
- The paper is well-structured and easy to follow.
- As we move towards a more data-centric view of deep learning, analysis of the composition of a dataset and how to best create a new dataset is a welcome area of research. This paper is one in this new area and sets a relatively decent standard for what should be done.

**Weaknesses:**

- All of the experiments are based on using ImageNet as the pretraining dataset. This makes me uncertain of whether the result relates to ImageNet only, as there are particularities in relation to dataset acquisition, labelling, diversity between samples in a class etc. that makes ImageNet unique. Another pre-training dataset would be very welcome and I think perhaps a needed addition to this paper.
- The paper seems like it is inspired by What Makes ImageNet Good for Transfer Learning by Huh et al, 2016; and this paper beyond just studying the trade-off between intra/inter-class diversity, actually ended generating a theoretically and empirically grounded rule for dataset construction.

**Questions:**

- Why not add another pre-training dataset to the mix?
- What happens when pre-training distribution doesn't match well with downstream dataset?

**Limitations:**

- Analysis restricted to ImageNet pretraining.

---

> ### Author Rebuttal · Authors · 2023-08-09
>
> >W1/Q1: *All of the experiments are based on using ImageNet as the pretraining dataset. This makes me uncertain of whether the result relates to ImageNet only, as there are particularities in relation to dataset acquisition, labelling, diversity between samples in a class etc. that makes ImageNet unique. Another pre-training dataset would be very welcome and I think perhaps a needed addition to this paper. Why not add another pre-training dataset to the mix?*
>
>
> We added new experiment results regarding using another dataset (Places365) as the pre-training dataset, and we can still observe the diversity trade-off and the aligned optimal class-to-sample ratio. Please find the details (**Figure R1**) in the general rebuttal.
>
> >W2: *The paper seems like it is inspired by What Makes ImageNet Good for Transfer Learning by Huh et al, 2016; and this paper beyond just studying the trade-off between intra/inter-class diversity, actually ended generating a theoretically and empirically grounded rule for dataset construction.*
>
> The "What Makes ImageNet Good for Transfer Learning" paper (referred as [1])  empirically examined the importance of a wide range of pre-training data characteristics on downstream performance. In contrast, we provide in-depth study and theory regarding a specific aspect, the trade-off between intra/inter-class diversity. Besides, we come up with a theoretically and empirically grounded rule for dataset construction in addition to studying the diversity trade-off. That is, one could estimate the optimal class-to-sample
> ratio using a small N and then use it to predict the optimal number of classes for building a large pre-training dataset. We do have a paragraph in the related work section to discuss the connection of [1] and our study.
>
> [1] What Makes ImageNet Good for Transfer Learning. Huh et al, 2016.
>
> >Q2: *What happens when pre-training distribution doesn't match well with downstream dataset?*
>
> In the general rebuttal (**Figure R3**), we added results of using dataset from painting domain (DomainNet-painting) as the downstream task (while the pre-training dataset used can be seen as from real image domain). We can see that although the two domains have different distributions of the data samples, we still observe the diversity trade-off and the aligned optimal class-to-sample ratio.

---

> > ### Comment · Reviewer_Ypk9 · 2023-08-18
> > **Response**
> >
> > Q1/W1. Thank you. Much appreciated. Concern is resolved.
> > W2. I think I listed this as a weakness in so far as it is work that was done before, however not in the depth here. I do agree that given the depth that the paper goes to it makes additional novel contributions.
> > Q2. I think something more out of distribution would have been more appreciated (maybe like earth observation imagery or biological images), but I see that the results are decent in this slightly different domain. If the paper is published, I think this would make the paper more convincing.
> >
> > Based on rebuttal I will raise my score to a 6.

---

> > > ### Author Response · Authors · 2023-08-21
> > > **Thank you for raising the score!**
> > >
> > > We would like to thank the reviewer for the comment and for raising the score. We will make sure that the additional experiments are included in the revision.

---

### Official Review · Reviewer_xyoX · 2023-07-06

**Soundness:** 3 good
**Presentation:** 3 good
**Contribution:** 3 good
**Rating:** 6
**Confidence:** 2

**Summary:**

This work studied the impact of the trade-off between the intra-class diversity (the number of samples per class) and the inter-class diversity (the number of classes) of a supervised pre-training dataset. The authors found that with the size of the pre-training dataset fixed, the best downstream performance comes with a balance on the intra-/inter-class diversity and show that the downstream performance depends monotonically on both types of diversity theoretically.

**Strengths:**

* The claims over how intra-class diversity and the inter-class diversity affect downstream tasks' performance were well supported by the empirical results, i.e., Fig. 1.

* The authors provided a theory on the impact and verified the theoretical findings via empirical results.

**Weaknesses:**

* The study was limited to fine-tuning on classification tasks only, other types of tasks, such as detection and segmentation, were not covered.

*  Though ResNet-18 is a classic architecture, it became less popular recently. It remains unclear if the same conclusion holds for other models (e.g., ViT) empirically.

**Questions:**

* Can authors elaborate line 195: "obtain the label $y_i$ for each $x_i$ by performing clustering.", for example, which clustering method was used? what is the input/feature for clustering algorithm?

---

> ### Author Rebuttal · Authors · 2023-08-09
>
> >W1:  *The study was limited to fine-tuning on classification tasks only, other types of tasks, such as detection and segmentation, were not covered.*
>
> This is a good point, our current study(theory) focuses on(supports) classification tasks and we would like to leave the exploration of other types of downstream tasks to the future work. We would update the paper to point out these directions of future work.
>
> >W2:   *Though ResNet-18 is a classic architecture, it became less popular recently. It remains unclear if the same conclusion holds for other models (e.g., ViT) empirically.*
>
> We added new experiment results regarding the ViT model (please find the **Figure R2 in the general rebuttal**) and our main findings still hold for the ViT model.
>
> >Q1: *Can authors elaborate line 195: "obtain the label y_i for each x_i by performing clustering.", for example, which clustering method was used? what is the input/feature for clustering algorithm?*
>
> Thanks for asking and we clarify the clustering method below both experimentally and theoretically below.
>
> **Experimentally**: We performed the K-means clustering algorithm on the embeddings of the 1,000 labels of ImageNet, where the label embeddings are obtained from the ConceptNet database (it includes label embeddings corresponding to the labels of ImageNet). Then the samples in ImageNet can be split into K clusters based on the obtained clusters of labels.  We will add more details about clustering to the revised version.
>
> **Theoretically**: Our theoretical result does not depend on the type of clustering method employed, and can apply to standard clustering methods such as K-means clustering and spectral clustering.

---

### Author Rebuttal · Authors · 2023-08-09

We thank all of the reviewers for their thoughtful feedback. Based on reviewers’ feedback and questions, we add several experiments and the results can be found in the attached PDF. In particular, we plot the curves of test error rate on downstream tasks across different class-to-sample ratio (similar to the Figure 2 of the main paper) for the following settings:

- **Figure R1: using the Places365 [1] dataset as the pre-training dataset.**  Based on reviewer Ypk9 and AXB6’s suggestion for experiments on an additional pr-etraining dataset, we pre-trained ResNet-18 on a new pre-training dataset, Places365, and tested the pre-trained models on downstream tasks;

- **Figure R2: using the ViT model.** Based on reviewer xyoX and AXB6’s comment on experimenting with other model like ViT, we pre-trained ViT-b-16 on the ImageNet dataset to show the diversity trade-off for models other than ResNet;

- **Figure R3: using larger/other domain datasets as downstream tasks.** Reviewer xyoX and AXB6 requested for experiments on other downstream tasks, so we tested ResNet-18 pretrained on ImageNet on two new downstream tasks from the DomainNet benchmark [2]: DomainNet-real and DomainNet-painting. They are from real and painting domains respectively and both have 345 categories. The DomainNet-real has >170K images and the DomainNet-painting has >70K images.

We focus on the test error rate across the class-to-sample ratio figure because it can demonstrate the trade-off of the intra-/inter-class diversity and show that the optimal class-to-sample ratio of different Ns are aligned, which is the main findings of this paper. From the figures, we can see that 1) overall, the largest and smallest class-to-sample ratio would not lead to the best test error rate, which indicates the diversity trade-off exists for these different settings; and 2) the best class-to-sample ratio for different Ns in the same setting are generally aligned, which reinforces our main findings in the paper.

[1] Places365 dataset: http://places2.csail.mit.edu/

[2] DomainNet benchmark: http://ai.bu.edu/M3SDA/

---

### Decision · Program_Chairs · 2023-09-21

**Decision:**

Accept (poster)

**Comment:**

The paper performs an analysis of the class-to-sample ratio in pretraining datasets. They propose a theoretical analysis based on generalization error bounds, as well as experiments, that conclude that the optimal class-to-sample ratio might remain constant across pre-training dataset sizes.

There is no clear consensus among reviewers on the exact novelty of the paper, but the main weaknesses -- single dataset, small model, and small downstream datasets -- have been partly cleared by the rebuttal and the reviewers agree that the in-depth theoretical analysis is new.

The paper is borderline but I feel the topic is important and timely. I accept the paper assuming there is a chance this type of analysis, joining theory and experiments, can be broadened and deepened and yield actionable insights at larger scale.